# Clinical Applications of Short Non-Coding RNA-Based Therapies in the Era of Precision Medicine

**DOI:** 10.3390/cancers14061588

**Published:** 2022-03-21

**Authors:** Ellen S. Smith, Eric Whitty, Byunghee Yoo, Anna Moore, Lorenzo F. Sempere, Zdravka Medarova

**Affiliations:** 1Department of Biochemistry, Northeastern University, Boston, MA 02115, USA; smith.ell@northeastern.edu; 2Athinoula A. Martinos Center for Biomedical Imaging, Massachusetts General Hospital and Harvard Medical School, Boston, MA 02129, USA; epwhitty@wpi.edu (E.W.); byoo@mgh.harvard.edu (B.Y.); 3Precision Health Program, Michigan State University, East Lansing, MI 48824, USA; moorea57@msu.edu; 4Department of Radiology, College of Human Medicine, Michigan State University, East Lansing, MI 48824, USA; 5Transcode Therapeutics, Inc., Boston, MA 02109, USA

**Keywords:** RNA interferences, microRNA, siRNA, drug delivery, precision oncology, nanomedicine, nanoparticle

## Abstract

**Simple Summary:**

RNA-based drugs are an attractive approach for personalized treatment of cancer and other diseases. This review focuses on two related classes of short non-coding RNA: microRNAs (miRNAs) and small interfering RNAs (siRNAs). miRNAs are endogenous short RNAs that bind multiple messenger RNAs (mRNAs) and prevent the production of their gene-products, whereas siRNAs are exogenous RNAs that target a single and specific mRNA for degradation. This review describes the development, challenges, and clinical successes of short RNA-based drugs. We provide several examples of how these RNA drugs are designed, chemically modified and delivered for treatment of different cancer types, cardiovascular disease, and rare genetic disorders. We highlight the similarities, differences, and considerations to maximize the treatment efficacy of miRNA-based vs. siRNA-based drugs.

**Abstract:**

Traditional targeted therapeutic agents have relied on small synthetic molecules or large proteins, such as monoclonal antibodies. These agents leave a lot of therapeutic targets undruggable because of the lack or inaccessibility of active sites and/or pockets in their three-dimensional structure that can be chemically engaged. RNA presents an attractive, transformative opportunity to reach any genetic target with therapeutic intent. RNA therapeutic design is amenable to modularity and tunability and is based on a computational blueprint presented by the genetic code. Here, we will focus on short non-coding RNAs (sncRNAs) as a promising therapeutic modality because of their potency and versatility. We review recent progress towards clinical application of small interfering RNAs (siRNAs) for single-target therapy and microRNA (miRNA) activity modulators for multi-target therapy. siRNAs derive their potency from the fact that the underlying RNA interference (RNAi) mechanism is catalytic and reliant on post-transcriptional mRNA degradation. Therapeutic siRNAs can be designed against virtually any mRNA sequence in the transcriptome and specifically target a disease-causing mRNA variant. Two main classes of microRNA activity modulators exist to increase (miRNA mimics) or decrease (anti-miRNA inhibitors) the function of a specific microRNA. Since a single microRNA regulates the expression of multiple target genes, a miRNA activity modulator can have a more profound effect on global gene expression and protein output than siRNAs do. Both types of sncRNA-based drugs have been investigated in clinical trials and some siRNAs have already been granted FDA approval for the treatment of genetic, cardiometabolic, and infectious diseases. Here, we detail clinical results using siRNA and miRNA therapeutics and present an outlook for the potential of these sncRNAs in medicine.

## 1. Introduction

Since its discovery by Drs. Andrew Fire and Craig Mello in 1998 [1,2], RNA interference (RNAi) has yielded a wealth of diagnostic and therapeutic targets. Here, we focus on the therapeutic potential of RNA-based short non-coding RNAs (sncRNAs) and highlight examples of small interfering RNA (siRNAs) and microRNA (miRNAs) therapies that have either entered clinical trial evaluation or have received approval for clinical use. sncRNAs are a promising therapeutic modality because of their potency and versatility [3,4,5,6,7]. Two representative classes of sncRNA-based therapeutics are siRNAs for single-target therapy and miRNA activity modulators for multi-target therapy (Figure 1B, [3,4,5,6,7]). siRNAs are exogenous double-stranded RNAs (dsRNAs) with natural or chemically modified nucleotides (Figure 2B) that target and degrade a specific and unique target RNA [7]. The high specificity of siRNA-mediated RNAi positions this approach at the core of personalized medicine [8,9]. siRNAs also have the capability for potent regulation of protein expression, a property that they derive from the fact that the underlying RNAi mechanism is catalytic and reliant on post-transcriptional mRNA degradation.

Mechanistically, siRNAs target mRNA for degradation and significantly decrease the biosynthesis of the cognate protein. Most commonly, siRNAs are RNA duplexes of ~21-nt in length. While RNAi can be accomplished by the administration of natural dsRNA oligonucleotides of various lengths, the injection of long dsRNA triggers an interferon response in mammalian cells and demonstrates limited processing into ~21 nucleotide-long siRNAs. This step is critical for guiding the RNAi cellular machinery (Figure 1A)—the RNA-induced silencing complex (RISC)—to silence gene expression [10,11,12].

Therapeutic siRNAs can be designed against virtually any RNA sequence in the transcriptome and target disease-causing mRNA variants. This capability makes siRNA-mediated RNAi both highly specific and broadly applicable to a variety of targets. An alternative approach uses synthetic single-stranded RNA (ssRNA) oligonucleotides such as antisense oligonucleotides (ASOs). The ASO can be a complementary sequence to either a protein-coding mRNA or non-coding RNA transcript and can induce gene silencing by forming a thermodynamically stable complex with the target RNA sequence [13,14,15]. However, ASOs lack the catalytic potency of siRNA and require a higher concentration of oligonucleotide to achieve a measurable therapeutic effect.

In contrast to siRNAs, miRNA activity modulators can be synthetic molecules that mimic the mature sequence of a particular miRNA, thus increasing its abundance in the cell or ASOs, known as antagomirs, that bind to a specific miRNA and decrease its activity by preventing its interaction with its targets. Accordingly, the increase or decrease of miRNA activity leads to changes in the expression of tens to hundreds of direct target genes and can have a more profound effect on global gene expression and protein output than siRNAs [16,17,18]. Endogenous miRNA precursors and exogenous miRNA mimics or siRNAs are processed by the same cellular machinery, which may include cleavage by RNAse III DICER and loading in the ARGONAUTE-containing miRISC/RISC (Figure 1A, [16,17]). Due to this enzymatic processing, chemical modifications of miRNA mimics are somewhat restricted compared to anti-miRNA ASOs [6,7]. After cell penetration or import and escape from endocytic routes, anti-miRNA ASOs bind, without any enzyme involvement, to perfectly complementary mature miRNAs, sequestering them away and preventing interaction with miRNA recognition sites on the target mRNAs.

Both miRNA activity modulators and siRNAs have significant clinical potential to solve the challenges of currently untreatable or poorly treatable conditions and aid in resolving many global health crises. These types of sncRNA-based drugs have been investigated in clinical trials, and some siRNAs have already been granted FDA approval (Figure 2A). As of November 2021, a search using the keyword “siRNA” as an interventional drug lists 89 clinical trials in the US National Library of Medicine (www.clinicaltrials.gov, accessed on 15 March 2022), of which 34 studies are actively recruiting or planning to recruit patients, 36 studies were successfully completed, and 12 were terminated or withdrawn (Table 1). Similarly, there are 10 clinical studies resulting from the search of “miRNA” as an interventional drug, of which 1 study is actively recruiting or planning to recruit patients, 5 were successfully completed, and 4 were terminated or withdrawn (Table 2). This seemingly lower success rate of miRNA-based drugs may reflect their more ambitious applications for treating cancer, whereas siRNAs may have been more conservatively selected to treat rare genetic diseases, metabolic and/or degenerative diseases. The major concern for these RNA drugs is off-target effects caused by the complexity of gene interactions, but the incidence rate of adverse effects is similar to that shown by other therapies [6,9]. Here, we focus our discussion on the results of clinical trials and present an outlook for the potential clinical uses of these sncRNA-based drugs. The array of RNA therapeutics that have shown promise preclinically, although not discussed in detail here, is immense and, if successful clinically, is undoubtedly poised to transform the way we treat diseases.

## 2. Delivery Strategies and Chemical Modifications to Reach the Target Organ or Tissues

Despite the promise of RNAi and RNA therapies, the most successful attempts have generally focused on hepatic targets and liver-involved conditions because of the availability of delivery vehicles (Table 1). To guarantee safe delivery into the desired loci, these RNA therapeutics, especially siRNAs and miRNA mimics, are often incorporated into nanoparticles or conjugated with other delivery agents (Figure 1C and Figure 2A, [3,5,6,21,22,23,24]). These include encapsulation in lipid nanoparticles (LNPs) whose delivery is governed by apolipoprotein E (ApoE) trafficking and/or conjugation to *N*-acetylgalactosamine (GalNAc), which reaches the liver by being endocytosed through the asialoglycoprotein receptor on hepatocytes. These delivery strategies are behind several therapies for hepatic genetic diseases and cardiometabolic disorders. Specific examples of the conditions that have been treated successfully with RNAi include hereditary ATTR amyloidosis, acute hepatic porphyria, primary hyperoxaluria, and alpha-1 antitrypsin deficiency, as detailed below. LNP encapsulation, GalNAc-conjugated (Alnylam Pharmaceuticals), and/or Dicerna’s GalXC™ GalNac-conjugated (Dicerna Pharmaceuticals) siRNAs platforms are behind many of these notable achievements (Figure 1C, [5,7,24]). For example, inclisiran, a GalNac-conjugated siRNA against *PCSK9* mRNA, has shown impressive efficacy against atherosclerotic cardiovascular disease [25]. In clinical trials, the inclisiran treatment was effective in reducing LDL-C in patients with elevated LDL-C despite receiving maximally tolerated statin therapy [26]. Unconjugated locked nucleic acid (LNA)-modified anti-miR-122 ASOs have also shown promise (Figure 2A). Designed as a treatment against hepatitis C, anti-miR-122 ASOs have been successfully delivered to the liver in non-human primates and humans in clinical trials [27] and have been shown to affect liver metabolism and lower low-density lipoprotein (LDL) cholesterol [27].

More recently, RNAi has successfully been used for the treatment of infectious diseases. These include Dicerna’s GalXC-modified RG6346 siRNA against the hepatitis B virus (HBV), Alnylam’s ALN-HBV02 siRNA against HBV, and Alnylam’s ALN-COV against severe acute respiratory syndrome coronavirus 2 (SARS-CoV-2) for treatment of coronavirus disease 2019 (COVID-19), reviewed in [28].

In the oncology space, attempts to apply RNAi for therapy have been challenged by the limited delivery of oligos directly into the cancer cells via physiologically relevant routes (such as intravenous and subcutaneous injection). Being able to reach cancer cells is important because most primary drivers of carcinogenesis and metastasis are cancer-cell-intrinsic processes. There is a vast array of well-studied and highly validated therapeutic targets that are expressed by cancer cells, and the unique capability of RNAi to essentially silence any gene would render these valuable but undruggable therapeutic targets druggable. Similarly, there is a candidate list of cancer-associated miRNAs (e.g., miR-10b, -21, -34a, -155) as well as long non-coding RNAs such as HOTAIR and MALAT1 that have been prioritized for therapeutic development [3,6,29,30,31,32,33]. Other delivery efforts have been directed at targeting other cell types in the tumor microenvironment (TME), such as immune cells. For example, the therapeutic pipeline of Ionis Pharmaceuticals focuses on the subcutaneous injection of unconjugated synthetic nucleic acids that include several candidates with a biological impact on the TME. Danvatirsen targets *STAT3*, a modulator of the immune response, while ION736 targets *FOXP3* and ION537 *YAP1*, both of which affect immune cell function in the TME (https://www.ionispharma.com/ionis-innovation/pipeline, accessed on 15 March 2022).

An alternative approach for cancer therapy that extends beyond sncRNA-based drugs and will not be discussed in detail involves the development of cancer vaccines. These are mutation-incorporating or neoantigen-expressing mRNAs that are exogenously delivered to professional antigen-presenting cells in the TME. These mRNAs are delivered in lipoplex formulations (BioNTech) or LNPs (Moderna, mRNA-5671), identical to the breakthrough mRNA vaccines against COVID-19 [34].

A strategy aimed at circumventing systemic delivery issues is to apply the RNA therapy locally (Figure 1C). Local delivery approaches have been evaluated in phase 2 clinical trials [5,21], including intravitreal injection for the treatment of macular degeneration [35], intranasal nebulized suspension for the treatment of respiratory syncytial virus (RSV) infection [36], and intra-lesional injection for the treatment of pachyonychia congenita [37]. In addition, intracardiac injection of miR-21 activity modulators (both mimics and inhibitors) have been investigated in pig models of infarction [38,39]. In the context of cancer, local delivery approaches involving mainly intratumoral injection have been extensively studied in preclinical models [3,6].

## 3. Clinical Application of Short ncRNAs

### 3.1. Unconjugated sncRNAs

#### 3.1.1. Diabetic Macular Edema and Age-Related Macular Degeneration

The first clinical trials that utilized siRNA targeting vascular endothelial growth factor (VEGF) in humans were performed in patients with macular degeneration [40]. Exudative, or “wet” age-related macular degeneration (AMD), is the leading cause of severe vision impairment for Americans over 65 years old [18]. When drusen accumulate on the retina during dry AMD, the resulting pressure on the retinal pigment epithelium causes an inflammatory response that upregulates VEGF, leading to choroidal neovascularization [18]. Diabetic macular edema (DME is) the leading cause of blindness occurring between the ages of 20 and 74; it can occur when upregulated VEGF causes increased permeability in the blood-retinal barrier, which leads to excess fluid in the eye, resulting in edema [41]. Prior clinical studies have established VEGF as an effective target for therapies, most commonly antibodies, to decrease vision loss that results from DME or AMD. The standard of treatment is anti-VEGF antibodies such as ranibizumab, the intravitreal injections of which pose the risk of lens injury, intravitreal bleeding, endophthalmitis, and retinal tears due to the frequency of their administration every 4 to 6 weeks [18]. Bevasiranib (also known as Cand5) is a siRNA that targets VEGF mRNA [41].

The advantage of siRNA-based therapies rather than antibody-based therapies is that the upregulation of VEGF is due to mRNA stabilization rather than increased translation, and the use of siRNA theoretically allows downregulation rather than blocking the action [18]. Bevasiranib does show a definite anti-angiogenic effect, which takes about 6 weeks to develop from the start of treatment because existing VEGF mRNA is not fully eliminated. In this case, combination treatment with the anti-VEGF antibody could be most effective [18]. A phase 3 clinical trial was designed to address the benefits of this combination treatment but was never started (NCT00557791). Multiple studies have indicated that bevasiranib’s main mechanism of action in the eye is not eliciting an RNAi response, but instead the RNA-mediated activation of the cell surface toll-like receptor 3 (TLR3), which suppresses CNV via intracellular signaling [40]. These siRNAs were not specifically designed for cell permeation, and the amounts of these siRNAs that reach their cognate target may have been lower than anticipated [40]. In 2009, a phase 3 clinical trial for bevasiranib was terminated because preliminary data suggested that the possibility of reaching the primary endpoint was very low (NCT00499590).

#### 3.1.2. Respiratory Syncytial Infection

Respiratory syncytial virus (RSV) is the leading cause of infant hospitalization in the US, partly because there is no vaccine, and very few therapeutic options are available against this infection [36]. For lung transplant patients, RSV infection is the most common community-acquired respiratory virus and is associated with bronchiolitis obliterans syndrome, which is a serious roadblock to patient and graft survival [42]. ALN-RSV01 is a siRNA developed by Alnylam Pharmaceuticals that targets the mRNA encoding the nucleocapsid protein, which is critical for RSV replication [36,43]. As is the case with lung-targeted siRNAs, delivery without a carrier works well as it can be applied directly to the mucosa and degraded by the nucleases if it enters the systemic circulation [44]. Two safety and tolerability studies with 101 healthy adults showed intranasal administration to be safe and well-tolerated, with doses of 150 mg either once or five times [43]. After 88 healthy adults were experimentally challenged with RSV, 71.4% of those who received the placebo were infected compared to only 44.2% of those who received ALN-RSV01 [36]. In Phase 2a trials with transplant patients naturally infected with RSV, ALN-RSV01 was found, in combination with standard of care, to reduce the incidence of new or progressive bronchiolitis obliterans syndrome (BOS). However, it did not meet the primary endpoint of reduced day 180 BOS and failed to progress to a phase 3 trial [42].

#### 3.1.3. Pachyonychia Congenita

Pachyonychia congenita (PC) is a dominant genetic condition characterized by thickened nails, leukokeratosis, keratoderma, and painful blistering primarily located on the soles of the feet [8]. Over 50% of patients are not able to walk without the aid of an ambulatory device [45]. Current treatments for PC are limited to inadequately effective symptom management through mechanical callus removal, topical keratolytics, and oral retinoids [45]. This condition results from mutations in keratins *K6a*, *K6b*, *K16*, or *K17*. A siRNA therapeutic, TD101, targets *K6a* mRNA, which is the most commonly mutated gene [8,45]. Quantitative reverse transcription PCR (qRT-PCR) was used to measure in vivo mRNA levels to verify the effectiveness of intradermal injection of TD101 in reducing the expression of mutant *K6a*. PC-10 cells and collected patient callus samples both expressed equal amounts of mutant and wild-type *K6a*. However, the mutant *K6a* expression was reduced by about 98% when TD101 was administered [8].

#### 3.1.4. Hepatitis C

miR-122, the most abundant hepatocellular miRNA, promotes hepatitis C virus (HCV) propagation. By binding to the 5′ end of HCV RNA, miR-122 protects this RNA from nuclease attack and masks an RNA motif that may elicit an innate immune response [46]. Chronic HCV can lead to cirrhosis and eventually hepatocellular carcinoma [47]. Miravirsen, which is an anti-miR-122 ASO (Figure 2A, Table 2) composed of locked nucleic acid (LNA) ribonucleotides that hybridize to mature miR-122 and block its interaction with HCV RNA, is currently in clinical trials [27]. LNAs have the second oxygen molecule linked to the 4′ carbon in the ribonucleotide (Figure 2B). This modification protects the oligonucleotide from nuclease degradation and can increase target affinity [47,48]. Treatment with miravirsen has been found to cause a dose-dependent decrease in viral load in chronic HCV patients in clinical trials, with no significant effects on the plasma levels of other miRNAs [46]. A placebo-controlled study of 5 weekly doses of miravirsen reduced plasma levels of miR-122 from 3.9 × 10^3^/4 to 3.1 × 10^1^/4 μL one week after the first dose in the experimental group and maintained these levels for the entirety of the study period in the highest-dose group [46]. In comparison, the mean plasma levels in the placebo group were 1.3 × 10^4^/4 μL at baseline and 1.1 × 10^4^/4 μL after one week of treatment. All dosed patients responded to therapy, with some having undetectable levels of miR-122 after treatment. Although the viral load decreased in dosed patients, there was no correlation between the decrease in miR-122 plasma levels and HCV viral load. Many of the patients with virological relapse after taking miravirsen had a C3U nucleotide change in the 5′UTR region of their HCV RNA, which is hypothesized to render this process miR-122-independent and therefore resistant to miravirsen [49,50]. miR-122 may also act as a tumor suppressor [51], which has raised some concerns that anti-miR-122 treatment could lead to an increased risk of hepatocellular carcinoma [51]. However, in preclinical studies, mice that were fed miravirsen for 5 weeks did not develop tumors. Still, this concern warrants further safety studies to evaluate the risk [27], given that mir-122-knockout mouse models do develop hepatocellular carcinoma [49,51].

#### 3.1.5. Acute Kidney Injury

Acute kidney injury (AKI) is a complicated condition characterized by a sudden decrease in glomerular filtration rate followed by an increase in serum creatinine concentration or oliguria. AKI generally occurs in the setting of acute or chronic illness. It affects approximately 20% of hospitalized patients. In clinical studies, the pooled incidence rate of AKI was 21.6%, with 10% of these patients requiring kidney replacement therapy [52,53]. QPI-1002 (Teprasiran, Quark Pharmaceuticals) is a siRNA targeting p53 for the prevention of AKI and delayed graft function after kidney replacement [54]. In a phase 2 clinical trial, 10 mg/kg of QPI-1002 reduced the incidence, severity, and duration of AKI after cardiac surgery in high-risk patients [54]. However, a phase 3 clinical trial was terminated early due to results not meeting efficacy outcomes at day 90 in these patients (NCT03510897).

#### 3.1.6. Alport’s Disease

miR-21 is a multi-faceted miRNA involved in carcinogenesis, fibrosis, inflammation, and immune response [6,30,55,56]. Alport syndrome is a genetic disorder caused by mutations in the genes encoding several α chains of collagen 4. Altered collagen 4 function compromises the capillary membranes in the kidney and other organs. miR-21 expression is upregulated in patients with Alport syndrome and genetic mouse models of this disease [57,58]. In the *Col4a3*^−/−^ mouse model, subcutaneous injection of 25 mg/kg anti-miR-21 ASO twice a week extended animal survival by 46% [57]. The anti-miR-21 ASO treatment significantly delayed glomerulosclerosis, the formation of glomerular crescents, and periglomerular fibrosis, which are associated with the progression of Alport syndrome [57]. Mechanistically, anti-miR-21 ASO treatment dampens TGF-β induced fibrosis and inflammation and protects PPARα/retinoid X receptor (PPARα/RXR)-dependent mitochondrial activity, which extends kidney function. In phase 1 clinical trials, patients with Alport syndrome received subcutaneous injections of 1.5 mg/kg of anti-miR-21 ASO (RG-021, now known as lademirsen) as a single dose or four doses given every 14 days (NCT03373786). Treatment was well-tolerated, and a phase 2 clinical trial is now actively recruiting patients with Alport syndrome to evaluate the therapeutic efficacy of lademirsen in maintaining kidney function (NCT02855268).

#### 3.1.7. Cardiovascular Disease

MRG-110, an LNA-modified ASO, targets miR-92a-3p as a therapy for cardiovascular disease and wound healing [48]. miR-92′s negative impact on wound healing due to anti-angiogenic effects, caused in part by the downregulation of pro-angiogenic integrin alpha 5, can be modulated by its inhibition, which is shown to improve vascularization after a heart attack, circulation after hind limb ischemia, and wound healing [48,59]. MRG-110 treatment causes the dose-dependent reduction of miR-92a-3p in whole blood. It also increases granulation tissue formation and promotes angiogenesis in experimental porcine and db/db mouse models of acute and chronic excision wounds [59]. These effects were greater in the MRG-110 group than in the positive control groups treated with rhVEGF-165 and rhPDGF-BB, indicating significant clinical promise. No safety concerns were of note. In human trials, it was found that with half-maximal dosing between 0.27 and 0.31 mg/kg, the effectiveness of the treatment was significant [48]. In the high dose groups, there was more than 95% inhibition after 24–72 h of treatment, and this inhibition remained significant for 2 weeks.

#### 3.1.8. Leukemias and Lymphomas

MRG-106 (cobomarsen), an LNA-modified ASO, targets miR-155 as a therapy for several hematologic cancers, including cutaneous T-cell lymphoma (CTCL), diffuse large B-cell lymphoma (DLBCL), and chronic lymphocytic leukemia (CLL). Functional studies and clinical data supported an important role of miR-155 in the etiology of mycosis fungoides (MF), the most common subtype of CTCL [60]. The way cobomarsen was formulated favored uptake by MF cells and CD4+ T-cells [60]. Cobomarsen treatment elevated expression of miR-155′s direct targets, BACH1, PICALM, and JARID2, and interfered with the pro-survival activity of miR-155 [60]. BACH1 (BTB and CNC homology 1, basic leucine zipper transcription factor 1) is a mediator of the oxidative stress response, PICALM (phosphatidylinositol binding clathrin assembly protein) is an endocytosis adaptor, and *JARID2* (jumonji and AT-rich interaction domain containing 2) is a negative regulator of leukemia cell proliferation [60]. A phase 1 clinical trial (NCT02580552) demonstrated the safety and low toxicity of cobomarsen in patients with hematological cancers. A phase 2 clinical trial (NCT03713320) was initiated in 2018 to determine the efficacy and safety of cobomarsen treatment in comparison to vorinostat, a histone deacetylase (HDAC) inhibitor, in patients with CTCL of the MF subtype. An appealing aspect of cobomarsen treatment is that it can be administered weekly in contrast to daily dosing of vorinostat; however, cobomarsen is administered intravenously vs. the oral delivery of vorinostat. This clinical trial was terminated after recruiting 37 patients due to business reasons and not any specific concerns with cobomarsen’s efficacy [61]. An anticipated crossover phase 2 clinical trial (NCT03837457) was terminated due to the lack of eligible patients. Genetic studies in *Mir-155*-knockout mouse models and successful treatment with anti-miR-155 ASO or similar inhibitors in in vivo animal models [6] and exceptional response in a single patient diagnosed with an aggressive subtype of DLBCL [62] support the further clinical evaluation of cobomarsen.

### 3.2. GalNAc-Conjugated sncRNAs

#### 3.2.1. Porphyria

Acute hepatic porphyria arises from defects in the enzymes essential to the hepatic heme biosynthesis pathways, leading to the accumulation of toxic heme intermediates [63]. In 2019, givosiran (GIVLAARI^®^, Alnylam Pharmaceuticals) was approved by the FDA as a first-of-its-kind siRNA for the treatment of acute hepatic porphyria [64,65]. Givosiran targets and degrades delta-aminolaevulinic acid-synthase 1 (*ALAS1*) mRNA, ameliorating acute porphyric attacks. Givosiran is a GalNAc-conjugated siRNA that targets *ALAS1* in the liver (Figure 2A). In phase 3 trials (Table 1), patients dosed subcutaneously with 2.5 mg/kg of givosiran monthly for 6 months had a 74% lower annual rate of porphyria attacks than patients who received the placebo [66]. The rate was 3.2 for the experimental group versus 12.5 for the placebo group. Notably, the experimental group had higher rates of renal and hepatic adverse events than the placebo group.

#### 3.2.2. Alpha-1 Antitrypsin Deficiency

Alpha-1 antitrypsin deficiency is caused by a mutation in the AAT (SERPINA1), leading to the misfolding and polymerization of the AAT protein that promotes fibrosis and cirrhosis upon accumulation in hepatocytes, leading to pulmonary and liver disease [67]. There are very few available treatments for this condition, which leads to approximately 76 liver transplants every year in the United States [67]. ARC-AAT is a medicine consisting of a cholesterol-conjugated siRNA and a melittin-derived peptide conjugated to GalNAc formulated as the excipient, EX1, and is injected intravenously. With a maximum dose of 4 mg/kg for patients with AATD and healthy volunteers, the reductions of mutant AAT were 78.8% and 76.1%, respectively. Despite phase 1 trials having no SAEs and only minimal adverse events, clinical trials were terminated due to toxicity findings in primates when using the ARC excipient. Trials are now underway to develop the drug for subcutaneous injection, which does not require a delivery excipient.

#### 3.2.3. Primary Hyperoxaluria

Overproduction of oxalate can lead to precipitation of calcium oxalate crystals and cause severe, end-stage renal disease, eventually leading to systemic oxalosis [68]. A siRNA developed for the treatment of primary hyperoxaluria was designed to target the hydroxyacid oxidase 1 gene (*HAO1*) [69], which encodes glycolate oxidase, a key enzyme in oxalate synthesis. Lumasiran, also known as Oxlumo™ (Alnylam Pharmaceuticals), is a subcutaneously administered siRNA-targeting *HAO1* mRNA that received FDA approval in November of 2020 for treating primary hyperoxaluria type 1 (Figure 2A). A phase 1/2 placebo-controlled trial of lumasiran achieved normal ranges of urinary oxalate (UOx) in 83% of patients receiving 3 mg/kg monthly, with no discontinuations or drug-related SAEs [70]. The mean maximal reduction was 75%, which remained at 66% 28 days after the last dose. A phase 3 trial dosed patients at baseline and at months 1, 2, 3, and 6 [71]. The primary endpoint was the percent change in 24-h UOx excretion over the 6-month study period, and the experimental group exhibited a 64.5% reduction, with effects seen within one month from the beginning of the treatment. At month 6, in 84% of the patients in the experimental group, UOx output fell within the range of 1.5 times from normal, whereas none of the placebo group patients exhibited any change [72].

#### 3.2.4. Hemophilia

Fitusiran is a therapeutic siRNA against hemophilia that binds to and degrades the mRNA that encodes antithrombin [73]. Fitusiran, or ALN-AT3, is targeted to the asialoglycoprotein receptor in the liver via GalNAc conjugation. Prophylaxis for hemophilia is effective but invasive, so less than half of adults use it, whereas others use on-demand therapy; 35% of hemophilia A patients and between 3% and 5% of hemophilia B patients develop resistance to factor infusions, which can require less-effective bypass therapy [73,74]. Clinical trials have been able to achieve dose-dependent knockdown of AT with weekly or monthly subcutaneous injections [73]. This has been associated with increased thrombin generation and a reduction in annualized bleed rate for both hemophilia A and B. Of 30 patients with hemophilia, 48% were bleed-free and 67% were free of spontaneous bleeds. Two patients had dental procedures and two had surgeries while being treated, and fitusiran was found to minimize bleeding during those procedures. Although most patients only had mild injection site reactions, one patient had a fatal thrombosis resulting from using high doses of clotting factor VIII against trial guidance. No antibody development was seen. Monthly doses of 80 mg are being administered in ongoing phase 3 trials (Table 1).

#### 3.2.5. Hepatitis B

ARC-520 is a hepatitis B drug comprised of two siRNAs that target HBV’s viral mRNA and inhibit protein production [75]. ARC-520 uses a dynamic polyconjugates (DPC) delivery system with GalNAc for hepatocyte targeting and is able to target HBV mRNA transcripts to reduce hepatitis B surface antigen (HBsAg) [75]. The use of two siRNAs rather than one makes it less likely to allow for the selection of resistant mutants. Phase 1 trials used doses ranging from 0.01 to 4 mg/kg of intravenous ARC-520 on 54 healthy volunteers in combination with an antihistamine to avoid drug administration-related histamine release. Another study investigated the reduction of HBsAg in response to dosing with ARC-520 in combination with nucleos(t)ide analogs NUC [76]. A dose-dependent reduction from baseline was demonstrated in HBV e-antigen negative patients. There was a significant reduction of HBsAg in the experimental group compared to the placebo; however, this effect was not observed in the low-dose experimental group. DCR-HBVS is another therapeutic from Dicerna Pharmaceuticals in phase I trials designed to treat chronic hepatitis B [77].

#### 3.2.6. Cholesterol Metabolism and Atherosclerotic Cardiovascular Disease

The number one cause of death in the United States is cardiovascular disease (CVD), and atherosclerotic heart disease is the number one cause of CVD morbidity and mortality [78]. After a period of improved CVD health outcomes, United States death rates are on the rise again [78]. One of the most targetable factors in CVD is low-density lipoprotein cholesterol; LDL-C-lowering therapies have demonstrated improved survival and decreased risk of cardiovascular events. A determining factor in the plasma concentration of LDL-C is proprotein convertase subtilisin-kexin type 9 (PCSK9). Monoclonal antibodies that bind to PCSK9 have been found to effectively lower LDL-C levels by blocking protein interactions on the LDL-C receptor, thereby increasing its expression on hepatocytes. Treatments targeting *PCSK9* mRNA can also reduce symptoms of familial hypercholesterolemia [79]. Inclisiran (Novartis) is a *PCSK9*-targeting siRNA that is injected subcutaneously in the abdomen at the dose of 300 mg [80]. It is administered in addition to statins and requires infrequent dosing; most studies have dosing on days 1 and 90, with every 6 months thereafter [26]. The ORION 10 and 11 trials demonstrated 54.1 and 51.9 mg/dL loss of LDL cholesterol levels, respectively, when treated with inclisiran versus placebo. Inclisiran was also found to raise HDL cholesterol and lower total cholesterol, triglycerides, lipoprotein (a), and apolipoprotein B. As renal clearance is the main form of elimination, safety studies have been done on patients with either ASCVD or hypercholesterolemia and varying degrees of renal impairment [26]. Although renal clearance decreased as renal impairment increased, the LDL cholesterol level changes were similar across the groups, with the treatment well tolerated [78]. The total occurrence of cardiovascular events in both the placebo and experimental groups in efficacy trials was too small to draw conclusions about the efficacy in reducing adverse events [26]. Trials are currently underway to determine long term effects: the ORION-8 study will look at long term safety over 990 days and is expected to conclude in 2023, while the ORION-4 study will look at the number of occurrences of major adverse cardiovascular events in patients treated with inclisiran as compared to the placebo group over 5 years and is expected to conclude in 2024 [81,82]. Inclisiran is currently under consideration by the FDA and could become the first siRNA drug with a wide impact in the USA and around the world [25].

#### 3.2.7. Atypical Hemolytic Uremic Syndrome

Atypical hemolytic uremic syndrome is a rare, life-threatening condition that causes renal impairment, thrombocytopenia, and microangiogenic hemolytic anemia [83]. Cemdisiran is a siRNA-targeting *C5* mRNA that is being developed to replace eclizumab as the standard of care. This siRNA inhibits terminal complement pathway activity and prevents the formation of membrane attack complex on kidney endothelial cells. Phase 1 trials of this drug were discontinued due to lack of funding as a result of the COVID-19 pandemic.

### 3.3. Lipid Nanoparticle Therapies

#### 3.3.1. Transthyretin Amyloidosis

In 2018, patisiran (ONPATTRO^®^, Alnylam Pharmaceuticals) was approved by the FDA as the first-of-its-kind siRNA therapeutic for the treatment of polyneuropathy caused by hereditary ATTR amyloidosis [84,85,86]. Mutant transthyretin clusters into insoluble fibers, causing amyloidosis in the heart, kidneys, GI tract, and nerves. Transthyretin amyloidosis is a multisystemic disease causing neuropathy, such as sensorimotor difficulties and cardiomyopathy that can be fatal. Currently, the available treatments include liver transplantation and the administration of transthyretin tetramer stabilizers. Patisiran targets and degrades both mutant variant and wild-type transthyretin (*TTR*) mRNA, ameliorating the symptoms of this rare genetic disease [84]. The targeting siRNA is encapsulated in lipid nanoparticles targeted to the liver, which are administered intravenously at 0.3 mg/kg for 80 min every 3 weeks. Earlier trials had successfully demonstrated a dose-dependent reduction of circulating transthyretin and slowed disease progression with the administration of the drug [84]. For phase 3 trials (NCT03997383), 148 patients were given patisiran, while 77 were given a placebo. In the experimental group, the median reduction of transthyretin was 81% at the completion of the trial at 18 months [84]. There was a 56% improvement in the neuropathy impairment score mNIS+7 neurophysiologic test in the experimental group compared to the placebo group’s 4% improvement. Every identified secondary endpoint showed significant improvements in the experimental group compared to the placebo group, with no safety concerns.

Another *TTR*-targeting siRNA, revusiran, reached phase 3 trials (NCT02319005), but dosing was discontinued earlier than planned due to the differences in death rates between the placebo and experimental groups. A post hoc safety investigation determined that patients that died while on treatment were older than 75 years old and had a more advanced heart failure at baseline [87]. On-target effects of lowering TTR mRNA were comparable in patients that lived or died after revusiran treatment. There was no evidence that revusiran treatment per se contributed to this increase in death rates [87].

#### 3.3.2. Liver Fibrosis

Heat shock protein 47 is necessary for the deposition of hepatic collagen and forming collagen fibrils. It works with hepatic stellate cells (HSCs) to form procollagen, which causes fibrosis when it is cleaved into insoluble collagen [23]. BMS-986263 is a lipid nanoparticle injected intravenously, composed of cationic and anionic lipids targeted to HSCs via surface retinoid moieties that contains an anti-HSP47 siRNA. BMS-986263 is designed to treat hepatic fibrosis resulting from hepatitis C, cirrhosis from nonalcoholic steatohepatitis, or other forms of liver impairment [88,89]. The drug has been tested for immunogenicity and safety, with no negative effects found to date.

#### 3.3.3. Hepatocarcinoma and Liver Metastases

MRX34 is a 23 nucleotide-long double-stranded miR-34 mimic in a 110 nm liposomal nanoparticle, developed to treat patients with advanced solid tumors (Figure 2A, [1,90]). miR-34 is a tumor suppressor that can downregulate over 30 oncogenes (e.g., MET, MYC, PDGFR-α). miR-34 transcription is regulated by p53 and serves as an important mediator of p53-dependent tumor suppression [1,90]. In 2016, a phase 1 clinical trial (NCT01829971) found the maximum tolerated dose to be 110 mg/m^2^ for non-HCC tumors and 93 mg/m^2^ for HCC after being administered to 47 stage IV cancer patients twice weekly for 3 weeks in 4-week cycles [1]. This clinical trial was amended to include premedication with dexamethasone to reduce the toxicity and side effects triggered by MRX34. Despite this modification, dexamethasone was not sufficient to mitigate the inflammation and immune-mediated toxicity of MRX34. Unfortunately, due to five serious adverse events that resulted in the death of four participants, this clinical trial was terminated. Another phase I clinical trial (NCT02862145) was carefully designed for the treatment of melanoma patients with MRX34, but it was withdrawn due to the fallout from the other trial. Initial speculations suggested that the liposomal formulation may have contributed to these adverse events, but the use of the same formulation in other clinical trials without serious events makes this possibility less likely [90]. It appears that unanticipated on-target effects of MRX34 on immune cell activation could be driving the immune-mediated toxicity. Delivery of the drug to the tumors was confirmed, and target gene expression was found to have the expected dose-dependent response in white blood cells [90]. Of the 85 patients that received MRX34, 3 had a partial response and 16 showed stable disease. The recommended phase 2 dose was determined to be 70 mg/m^2^ for HCC and 93 mg/m^2^ for all other cancers. While a refined approach to target more selective cases with MRX34 may provide therapeutic benefits, the future clinical development of MRX34 is unclear [91].

#### 3.3.4. Pancreatic Cancer

Ninety percent of pancreatic cancers are pancreatic ductal adenocarcinoma (PDAC), and 80%–85% of pancreatic cancer cases are stage III or IV when diagnosed. Therefore, the vast majority of patients are not candidates for curative surgery [92]. Chemotherapy, most commonly gemcitabine, with or without radiation, is the current gold standard of care [93,94]. Ribonucleotide reductase (RR) is a good target for tumor treatment as it controls a rate-limiting step essential for DNA replication and can therefore reduce PDAC growth potential [95]. While the RR M1 subunit expression is constant, the M2 subunit is mainly expressed during DNA replication. In vivo RRM2 siRNA therapies resulted in the presence of an mRNA fragment that indicates that mRNA cleavage occurred at the site consistent with the RNAi mechanism [22]. siRNA targeting RRM2 has also been shown to inhibit gemcitabine resistance in pancreatic adenocarcinomas [96]. The RRM2 siRNA is contained within a linear cyclodextrin-based polymer and polyethylene glycol nanoparticle with human transferrin (hTf) targeting ligands designed to circulate before accumulating in tumors [22]. It has been demonstrated that targeting through these nanoparticles inhibits tumor growth more significantly than the same nanoparticle without transferrin targeting [97]. When biopsies from treated melanoma patients were collected, the amount of intracellular nanoparticles correlated directly with the delivered dose. This was the clinical demonstration of this observation [22].

Ninety percent of PDACs have a *KRAS* mutation, most commonly the G12D mutation. Preclinical PDAC models have shown that KRAS inhibition leads to improved survival, slowed migration, and increased tumor necrosis in mouse models [3,30,94]. Natural or synthetic exosomes, similar to chemically-defined lipid nanoparticles, can be used as encapsulation and delivery systems for siRNAs and miRNA activity modulators in PDAC animal models [3,30,98,99,100,101,102,103,104,105,106,107]. Several of these exosome-loaded strategies directly target mutant KRAS mRNA [101,107]. Engineered exosome-loaded siRNAs against KRAS G12D offered a higher therapeutic benefit than the same siRNAs loaded in liposomes in several KRAS^G12D^-driven orthotopic models [101]. The engineered exosomes can be produced at a large scale in bioreactors under the good manufacturing practice (GMP) standards that are required for clinical applications [107]. However, the first platform to reach the clinical stage relies on the local delivery of the *KRAS*-targeting siRNA. The LODER™ (Local Drug EluteR) platform consists of a poly(lactic-co-glycolic acid) (PLGA) matrix that slowly releases siRNAs (Figure 1C). The siG12D-LODER™ is a very small biodegradable implant that releases siRNA targeting the G12D and G12V mutations on the KRAS gene and can be implanted directly into pancreatic tumors during endoscope ultrasound biopsy procedures [92]. The siRNAs are released at a rate of about 1 mm/day from the inner core of the tumor for 4 months, with full saturation of the tumor occurring around one week after commencing the treatment. The polymeric PLGA matrix is made entirely of materials generally recognized as safe and allows for prolonged release without the degradation of the drug over time [92,108]. Twelve patients with PDAC treated with siG12D-LODER™ combined with chemotherapy (gemcitabine and/or FOLFIRINOX) received CT scans after treatment that showed no tumor progression, and two of them showed partial response (https://www.silenseed.com/wp-content/uploads/2020/05/Silenseed-Presentation-21-Jan-2020-FC.pdf, accessed on 15 March 2022). The dose was set at a maximum of 3 mg, above which RNAi saturation occurred. Any adverse events that occurred were attributed to either the procedure or the chemotherapy rather than the siRNA nanodrug. Notably, two of the patients were still alive 27 and 30 months after treatment, respectively.

### 3.4. DOPC Nanoliposomes

#### Solid Tumors

EphA2 is a receptor tyrosine kinase associated with poor outcomes that are thought to play a significant role in the proliferation, survival, and migration of tumor cells [109]. It is mainly expressed in epithelial cells and overexpressed in many cancers. EPHARNA DOPC (18:1 PC *cis* 1,2-dioleoyl-sn-glycero-3-phosphatidylcholine) is a neutral nanoliposomal delivery for EphA2-targeting siRNA. It was found to be well tolerated overall, both in mice and non-human primates (macaques). Phase 1 safety trials of EPHARNA DOPC in solid tumors are underway and not yet completed [110].

### 3.5. Mini-Cells, Larger Encapsulation

#### Mesothelioma

Malignant pleural mesothelioma (MPM) is an incurable pleura cancer related to exposure to asbestos that is associated with a 2- to 5-fold downregulation of the miR-15/107 group [111,112]. Chemotherapy can reduce tumor burden and prolong life, but other therapies are needed. MesomiR1 is a drug for MPM using TargomiR technology, which delivers RNA mimics, miR-15/107 in this case, to targeted bacteria cells as an alternative to liposomal or nanoparticle delivery. These EnGeneIC Delivery Vehicle (EDV)™ minicells are nonviable bacterial cells of about 400 nm in diameter, modified with antibodies on the surface for tumor cell targeting. In this first-in-human trial, MesomiR-1 was administered intravenously to patients with MPM at doses of 5 × 10^9^ TargomiRs over 20 min once weekly. The majority of patients had a short-term inflammatory response, lymphopenia, and neutrophilia after administration, and about half had short-term pain at the site of the cancer. After 8 weeks of the treatment described above, four patients had stable disease, one patient had improved, and one patient had progressive disease. The patient who improved significantly had, prior to treatment with MesomiR-1, undergone six rounds of chemotherapy with no response beyond the stabilization of disease. After treatment with the miRNA mimic, he showed evidence of partial response on computed tomography (CT) imaging and experienced a marked improvement in chest pain and respiratory function. This response occurred after 8 weeks of treatment, with the lowest dose administered in this trial.

## 4. miRNA Therapeutics in Preclinical Investigation

As of 2021, 3638 human miRNAs have been registered in the open miRNA database (miRDB) [113]. miRNA therapies such as mimics and anti-miRNA ASOs present a significant potential for cancer and other pathologies through their ability to modulate many target genes at once. In addition to the discussed miRNA-based therapeutics that have reached clinical trials, there is a long list of miRNAs that are under investigation in the preclinical setting for oncological applications. miR-710 has been found to be downregulated in metastatic lesions regardless of the location [114]. When the miR-710 mimic was applied to murine metastatic breast adenocarcinoma cells, it was able to inhibit their viability, migration, and stemness [114]. Enhanced expression of E-cadherin accompanied by reduced vimentin expression indicated a change in epithelial to mesenchymal markers, an important factor in metastasis formation. These results point to miR-710 as a potential drug target for preventing metastasis. The treatment of stage IV breast cancer is currently limited to quality-of-life care and needs more therapeutic options to prevent and treat metastasis [115]. The inhibition of miR-10b with MN-anti-miR10b, a therapeutic consisting of magnetic nanoparticles conjugated to LNA antimiRs, led to a significant reduction in mortality in animal models of metastatic breast cancer [115,116]. miR-29 family members have been found to be tumor-suppressing through their upregulation of p53, a tumor suppressor that is downregulated in a large majority of human cancers [117]. miR-29 also suppresses p85α and CDC43, both of which downregulate p53. Treatment of human cancer cells with miR-29 increased p53 two- to three-fold, depending on the member of the family, and induced apoptosis correlated to p53 reduction. An ASO was used to further illustrate this relationship by showing the inhibition of the miRNA suppressed activation of p53. Restoration of p53 in mouse models resulted in tumor regression so that the relationship between miR-29 and p53 shows therapeutic potential for miR-29 mimics in cancer therapies. miR-21 is frequently upregulated in solid and hematological tumors, but the predominant cell type of expression is variable among tumor types and within subtypes of a given cancer [6,30,55]. Generally, miR-21 is considered to have oncogenic activity [55]. When expressed in cancer cells, miR-21 promotes cell proliferation, migration, and invasion and is a negative regulator of apoptosis [55]. However, miR-21 is also expressed and active in other cell types of the TME [6,55]. When miR-21 is expressed in cancer associated-fibroblasts and subsets of immune cells, including macrophages and T-cells, its overall effect is more complex and/or can have mixed effects as a tumor-promoting or tumor-restraining factor in a cell type-specific manner [6,30,55,118,119]. Dysregulation of a core set of miR-21 target genes, including *PTEN*, *PDCD4*, *RECK*, *SMAD7*, and *STAT3* in different cell types of the TME, can lead to undesirable tumor-promoting effects, which makes it challenging to delineate a clear targeting strategy for cancer treatment [6,55]. Additionally, miR-21 plays important physiological roles in fibroblast and wound response [119,120], and there are concerns about interference with its activity in non-targeted organs or cell types.

## 5. Conclusions

RNA therapies have demonstrated clinical potential for both the treatment of cancer and other pathologies. Therapeutic delivery and resulting adverse events remain significant roadblocks in implementing many of these drugs into clinical practice, but the FDA approval of three Alnylam Pharmaceuticals’ siRNA therapies has been a milestone in developing therapies tailored to disease-driving target genes. While it seems that RNAs can be administered “naked” in closed-compartment organs such as eyes and lungs, more research is needed for systemic administration. Lipid nanoparticles represent a promising delivery method, but some challenges remain because of their potential to elicit an immune response, relatively low circulation times, and relatively large size. The use of GalNAc for the delivery and targeting of siRNAs has made significant progress, but delivery systems targeted to organs other than the liver would broaden the range of diseases that could be treated with RNA therapies. The LODER system, for example, showed promise for the treatment of solid tumors. miRNAs have also demonstrated clinical potential for the treatment of cancer and other pathologies. Still, a better understanding and ability to predict their on- and off-target effects in human subjects remains challenging despite rigorous safety and toxicity studies in animal models. In addition to the sncRNA-based therapies in clinical trials, there is a large list of RNA therapeutics that are under investigation in preclinical settings in research labs as well as in the pipeline of several pharmaceutical companies. Within this context, it is reasonable to expect that there will be more impactful RNA therapeutics in clinical trials in the next decade and that the new technologies and refinements will overcome some of the remaining challenges for broad clinical applications.

## Figures and Tables

**Figure 1 cancers-14-01588-f001:**
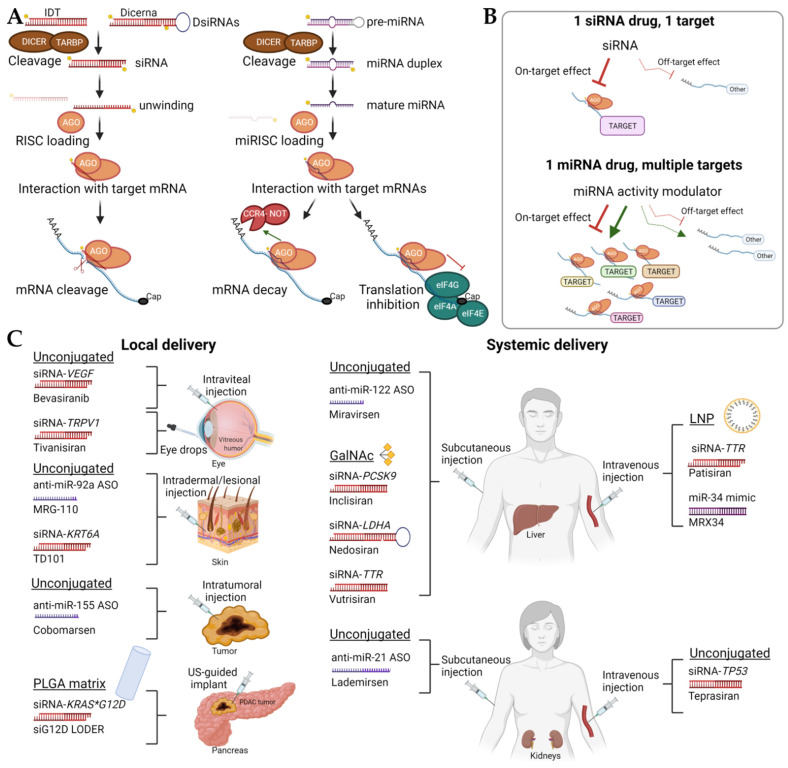
Processing, delivery strategies, and target engagement of RNA therapeutics. (**A**) miRNA precursor stem-loop hairpin and longer siRNA precursors (e.g., dicer substrate [DsiRNA] by IDT, or small hairpin RNAs such as the Dicerna nicked dsRNA stemloop) are processed by the DICER-containing complex. Transitory double-stranded product is similarly unwound and loaded into the ARGONAUTE-containing RNA-induced silencing complex (RISC). (**B**) While siRNAs are designed to specifically and perfectly match the complementary sequence of the cognate target mRNA, miRNAs bind to partially complementary sequences of multiple target mRNAs. (**C**) We provide representative examples of local and systemic delivery strategies to enhance the accumulation of the RNA therapeutics in the intended site of treatment. For systemic delivery, chemical modifications (shown in Figure 2B), encapsulation, and/or targeting moieties can facilitate retention by a specific organ or cell type. Abbreviations: AGO = argonaute RISC component; CCR4-NOT = carbon catabolite repression-negative on TATA-less complex; DICER = ribonuclease III Dicer1; Dicerna = Dicerna Pharmaceuticals; eiF4 = eukaryotic translation initiation factor 4; GalNAc = *N*-acetylgalactosamine; IDT = Integrated DNA Technologies; LDHA = hepatic lactate dehydrogenase A; LPN = lipid nanoparticle; *PCSK9* = proprotein convertase subtilisin/kexin type 9; PLGA = poly(lactic-co-glycolic acid); TARBP = TAR (HIV-1) RNA-binding protein 1; *TRPV1* = transient receptor potential cation channel subfamily V member 1; *TP53* = tumor protein p53; *TTR* = transthyretin; *VEGF* = vascular endothelial growth factor.

**Figure 2 cancers-14-01588-f002:**
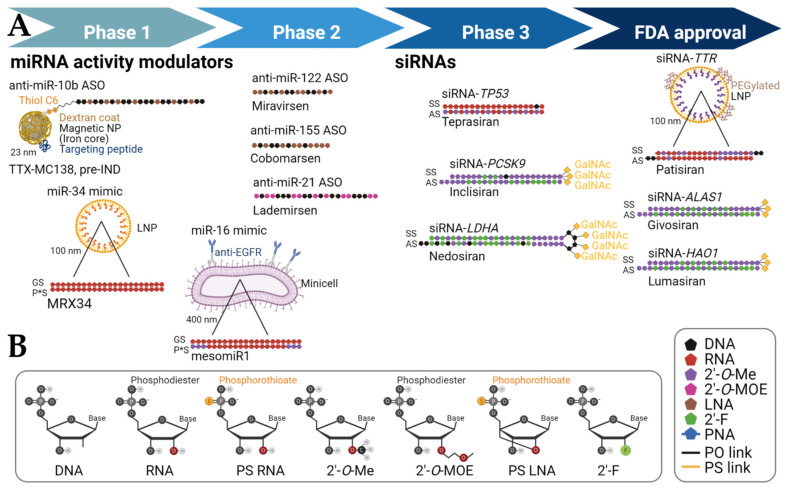
Chemical modifications and specific sequence patterns of these modifications facilitate the clinical application of RNA therapeutics. (**A**) Stage of clinical development of representative RNA therapies and chemical formulations behind their therapeutic effect. Pattern and location of chemical modifications are approximations (in some cases, the exact sequence is not disclosed). For siRNAs, the top strand (5′-end on the left) is the sense strand (SS), and the bottom strand (5′-end on the right) is the active antisense (AS) strand. For miRNA mimics, the top strand is the active mature miRNA guide (GS), and the bottom strand is the passenger strand (P*S). Nucleotides and other molecules are not drawn to scale. (**B**) Chemical structure of common sugar and backbone modifications of RNA therapeutics in clinical trials are depicted (inset). Abbreviations: 2′-F = 2′-deoxy-2′- fluoro; 2′-*O*-Me = 2′-*O*-methyl; 2′-*O*-MOE = 2′-*O*-methoxyethyl; *ALAS1* = delta-aminolaevulinic acid-synthase; ASO = antisense oligonucleotide; EGFR = epidermal growth factor receptor; GalNAc = *N*-acetylgalactosamine; *HAO1* = hydroxyacid oxidase 1; IND = investigational new drug; *LDHA* = hepatic lactate dehydrogenase A; LPN = lipid nanoparticle; NP = nanoparticle; *PCSK9* = proprotein convertase subtilisin/kexin type 9; PEG = polyethylene glycol; PS = phosphorothioate; PO = phosphodiester; *TP53* = tumor protein p53; *TTR* = transthyretin.

**Table 1 cancers-14-01588-t001:** RNAi-based therapies in phase 3 clinical trials.

Target Gene	Drug Name	Chemistry	Platform	Delivery	Treatment (Organ Site)	Sponsor	References
*ALAS1*	ALN-AS1 (Givosiran) *	siRNA (2′-*O*-Me, 2′F, partial PS backbone)	GalNAc conjugation, 2.5 mg/kg	Subcutaneous	Acute Hepatic Porphyrias (Liver)	Alnylam Pharmaceuticals (Cambridge, MA, USA)	NCT03338816, Completed
*AT*	Fitusiran ALN-AT3SC (Fitusiran)	siRNA (2′-*O*-Me, 2′-F, partial PS backbone)	GalNAc conjugation	Subcutaneous	Hemophilia A or B (Liver)	Genzyme, a Sanofi Company (Cambridge, MA, USA)	NCT03417102/03417245, Completed; NCT03754790/NCT03549871, Active
*CASP2*	QPI-1007	siRNA (2′-*O*-Me)	Up to 3 mg	Intraviteal	Acute Nonarteritic Anterior Ischemic Optic Neuropathy (Eye)	Quark Pharmaceuticals (Newark, CA, USA)	NCT02341560, Terminated
*HAO1*	ALN-GO1 (Lumasiran) *	siRNA (2′-*O*-Me, 2′F, partial PS backbone)	GalNAc conjugation, up to 3 mg/kg	Subcutaneous	Primary Hyperoxaluria Type 1 (Liver)	Alnylam Pharmaceuticals (Cambridge, MA, USA)	NCT03681184, Active; NCT03905694, Active; NCT04152200, Active
*LDHA*	DCR-PHXC (Nedosiran)	DsiRNA pseudo-hairpin (2′-*O*-Me, 2′F, DNA, partial PS backbone)	GalXC	Subcutaneous	Hyperoxaluria (Liver)	Dicerna Pharmaceuticals (Lexington, MA, USA)	NCT04042402, Enrolling by invitation
*PCSK9*	Inclisiran	siRNA (2′-*O*-Me, 2′F, internal DNA, partial PS backbone)	GalNAc conjugation, 300 mg	Subcutaneous	Homozygous Familial Hypercholesterolemia (Liver)	Novartis Pharmaceuticals (Basel, Switzerland)	NCT03851705, Active; NCT04659863, Recruiting
*PCSK9*	Inclisiran	siRNA (2′-*O*-Me, 2′F, internal DNA, partial PS)	GalNAc conjugation, 300 mg	Subcutaneous	Atherosclerotic Cardiovascular Disease (ASCVD) or ASCVD High Risk and Elevated LDL-C (Liver)	Novartis Pharmaceuticals (Basel, Switzerland)	NCT04765657, Recruiting
*PCSK9*	Inclisiran	siRNA (2′-*O*-Me, 2′F, internal DNA, partial PS backbone)	GalNAc conjugation, 300 mg	Subcutaneous	Prevent Cardiovascular events in Participants with Established Cardiovascular Disease (Liver)	Novartis Pharmaceuticals (Basel, Switzerland)	NCT05030428, Recruiting
*TP53*	QPI-1002 (Teprasiran)	siRNA (2′-*O*-Me)	-	Intravenous	Improved Graft Function after Donor Kidney Transplant (Kidney)	Quark Pharmaceuticals (Newark, CA, USA)	NCT02610296, Completed
*TP53*	QPI-1002 (Teprasiran)	siRNA (2′-*O*-Me)	-	Intravenous	Prevention of acute kidney injury after cardiac surgery (Kidney)	Quark Pharmaceuticals (Newark, CA, USA)	NCT03510897, Terminated
*TRPV1*	SYL1001 (Tivanisiran)	siRNA	Ophthalmic solution	Periocular	Sjögren′s Syndrome, Dry eye (Eye)	Sylentis, S.A. (Madrid, Spain)	NCT04819269, Recruiting
*TRPV1*	SYL1001 (Tivanisiran)	siRNA	Ophthalmic solution, 11.25 mg/mL	Periocular	Moderate to Severe Dry Eye Disease (Eye)	Sylentis, S.A. (Madrid, Spain)	NCT03108664, Completed
*TTR*	ALN-TTR02 (patisiran)*	siRNA (2′-*O*-Me, DNA overhangs)	Lipid nanoparticle	Intravenous	Transthyretin-Mediated Polyneuropathy (Liver)	Alnylam Pharmaceuticals (Cambridge, MA, USA)	NCT01960348, Completed
*TTR*	ALN-TTR02 (patisiran)	siRNA (2′-*O*-Me, DNA overhangs)	Lipid nanoparticle, 0.3 mg/kg	Intravenous	hATTR amyloidosis with disease progression after liver transplant (Liver)	Alnylam Pharmaceuticals (Cambridge, MA, USA)	NCT03862807, Completed
*TTR*	ALN-TTR02 (patisiran)	siRNA (2′-*O*-Me, DNA overhangs)	Lipid nanoparticle	Intravenous	ATTR Amyloidosis with Cardiomyopathy (Liver)	Alnylam Pharmaceuticals (Cambridge, MA, USA)	NCT03997383, Active
*TTR*	ALN-TTRSC (Revusiran)	siRNA (2′-*O*-Me, 2′-F)	GalNAc conjugation	Subcutaneous	Transthyretin-Mediated Familial Amyloidotic Cardiomyopathy (Liver)	Alnylam Pharmaceuticals (Cambridge, MA, USA)	NCT02319005, Completed
*TTR*	ALN-TTRSC02 (Vutrisiran)	siRNA (2′-*O*-Me, 2′-F, partial PS backbone)	GalNAc conjugation, 25 mg	Subcutaneous	Transthyretin Amyloidosis with Cardiomyopathy (Liver)	Alnylam Pharmaceuticals (Cambridge, MA, USA)	NCT04153149, Active
*TTR*	ALN-TTRSC02 (Vutrisiran)	siRNA (2′-*O*-Me, 2′-F, partial PS backbone)	GalNAc conjugation	Subcutaneous	hATTR Amyloidosis (Liver)	Alnylam Pharmaceuticals (Cambridge, MA, USA)	NCT03759379, Active
*VEGF*	Bevasiranib	siRNA	Up to 2.5 mg	Intraviteal	Age-Related Macular Degeneration following initiation of anti-VEGF Lucentis^®^ antibody therapy (Eye)	OPKO Health, Inc. (Miami, FL, USA)	NCT00557791, Withdrawn

This list includes only Phase 3 clinical trials that resulted from the search of the keyword “siRNA” as an interventional drug in the US National Library of Medicine (www.clinicaltrials.gov, accessed on 15 March 2022). * Please note that this siRNA drug has since received FDA approval. Abbreviations: 2′-*O*-Me = 2′-*O*-methyl; 2′-F = 2′-deoxy-2′-fluoro; *ALAS1* = delta-aminolaevulinic acid-synthase; *AT* = antithrombin; *CASP2* = caspase-2; DsiRNA = Dicer substrate siRNA; GalNAc = *N*-acetylgalactosamine; *HAO1* = hydroxyacid oxidase 1; hATTR = hereditary transthyretin-mediated amyloidosis; *LDHA* = hepatic lactate dehydrogenase A; LDL-C = low density lipoprotein cholesterol; *PCSK9* = proprotein convertase subtilisin/kexin type 9; PS = phosphorothioate; *TRPV1* = transient receptor potential cation channel subfamily V member 1; *TP53* = tumor protein p53; *TTR* = transthyretin; *VEGF* = vascular endothelial growth factor.

**Table 2 cancers-14-01588-t002:** miRNA-based therapies in clinical trials.

miRNA Modulation	Drug Name	Chemistry	Platform	Delivery	Disease (Organ Site)	Sponsor	Clinical Status	References
miR-10b inhibition	RGLS5579	ASO (2′-O-MOE, partial PS backbone)	-	Intravenous or intracranial	Glioblastoma (Brain)	Regulus Therapeutics (San Diego, CA, USA)	Pre-IND filing	[19]
miR-10b inhibition	TTX-MC138	ASO (partial LNA, partial PS backbone)	Dextran-coated iron oxide magnetic nanoparticle	Intravenous	Metastatic breast cancer (Lung, other organs)	Transcode Therapeutics (Boston, MA, USA)	Pre-IND filing, scheduled 2022	[20]
miR-16 restoration	mesomiR1 (TargomiR)	dsRNA mimic (2′-O-Me on passenger strand only)	Bacterial minicells with anti-EGFR bispecific antibody	Intravenous	Recurrent malignant pleural mesothelioma and non-small cell lung cancer (Lung)	Asbestos Diseases Research Foundation (New South Wales, Australia), EnGeneIC Limited (Lane Cave West, Australia)	Phase 1	NCT02369198, Competed
miR-21 inhibition	Lademirsen (SAR339375; previously known as RG-012 [Regulus])	ASO (sugar 2′ position modifications, PS backbone)	Unconjugated	Subcutaneous, 1.5 mg/kg	Alport syndrome (Kidney)	Genzyme, a Sanofi Company (Cambridge, MA, USA)	Phase 1	NCT02855268, Completed
miR-21 inhibition	Lademirsen (SAR339375; previously known as RG-012 [Regulus])	ASO (sugar 2′ position modifications, PS backbone)	Unconjugated	Subcutaneous	Alport syndrome (Kidney)	Genzyme, a Sanofi Company (Cambridge, MA, USA)	Phase 2	NCT02855268, Recruiting
miR-34a restoration	MRX34	dsRNA mimic	Liposome	Intravenous	Primary liver cancer or other selected solid tumors or hematologic malignancies (Liver, other organs)	Mirna Therapeutics (Austin, TX, USA)	Phase 1	NCT01829971, Terminated; NCT02862145, Withdrawn
miR-92a inhibition	MRG-110	ASO (LNA-modified)	-	Intradermal	Wound healing	miRagen Therapeutics, Inc. (Boulder, CO, USA)	Phase 1	NCT03603431, Completed
miR-122 inhibition	Miravirsen (SPC3649)	ASO (partial LNA, PS backbone)	Unconjugated	Subcutaneous	HCV chronic infection (Liver)	Copenhagen, Denmark	Phase 2	NCT01200420, Completed
miR-155 inhibition	MRG-106 (Cobomarsen)	ASO (partial LNA)	Unconjugated	Intratumoral and/or intravenous or subcutaneous	Certain lymphomas and leukemias, including CTCL [mycosis fungoides subtype], CLL, DLBCL [activated B-cell (ABC) subtype], and ATLL	miRagen Therapeutics, Inc. (Boulder, CO, USA)	Phase 1	NCT02580552, Completed
miR-155 inhibition	MRG-106 (Cobomarsen)	ASO (partial LNA)	Unconjugated	Intravenous	CTCL [mycosis fungoides subtype]	miRagen Therapeutics, Inc. (Boulder, CO, USA)	Phase 2	NCT03713320 and NCT03837457, Terminated

This list includes all studies that resulted from the search of the keyword “miRNA” as an interventional drug in the US National Library of Medicine (www.clinicaltrials.gov, accessed on 15 March 2022). Additional examples are included when there is strong evidence for clinical evaluation. Abbreviations: 2′-*O*-Me= 2′-*O*-methyl; 2′-*O*-MOE = 2′-*O*-methoxyethyl; ASO = antisense oligonucleotide; ATLL = adult T-cell leukemia/lymphoma; CLL = chronic lymphocytic leukemia; CTCL = cutaneous T-cell lymphoma; DLBCL = diffuse large B-cell lymphoma; dsRNA = double-stranded RNA; LNA = locked nucleic acid; PS = phosphorothioate.

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
