# Peer review of "Clinical Applications of Short Non-Coding RNA-Based Therapies in the Era of Precision Medicine"

_cancers, 2022, doi:10.3390/cancers14061588_

Round 1
Reviewer 1 Report
In the paper “Clinical applications of RNA-based therapies in the era of precision medicine” the authors show a review of current uses of siRNAs and/or miRNAs for clinical therapy.
RNA-based strategies for gene regulation are gaining interest and they have become a trending subject matter in the last years. Their use in molecular medicine research has provided a rationale for the therapeutical treatment of several genetic disorders.
As a consequence, a great number of reviews dealing with this topic have been recently published and it seems difficult to contribute to new advances in this field. Many of those reviews have been properly referenced by the authors and are focused in miRNA-based (3,6) or siRNA-based (22) therapies.
Fortunately, the authors have been able to detect and exploit gaps in the literature so they can offer a new viewpoint: they integrate both miRNA and siRNA-based therapies and they focus on those that have entered clinical trial evaluation or are FDA-approved.
Furthermore, the writing is ordered, clear and easy to understand. I am glad to observe that not only satisfactory results are presented but also those that have failed because they do not meet the expectations or those that have been prematurely terminated due to technical or medical incidences.
However, I would like to point out some questions that could be taken into consideration before publication:
- Increase the accuracy of the title: RNA-based therapies not only rely on short non-coding RNAs, but also in mRNAs, long non-coding RNAs, circular RNAs or ribozymes (see https://doi.org/10.1080/15476286.2022.2027150). It would be more appropriate to use “clinical applications of therapies based on sncRNAs…”, because the current title do not fulfill the expectances.
- Figure 1:
- What do “IDT”and “Dicerna” stand for? Are they the names of the companies that provide the dsRNAs?. It is confusing, as it seems to be a schematic representation of sRNA-mediated silencing in the cell…
- Abbreviations in the caption:
- Some of them are missing: LDHA
- Some of them do not appear in the figure: HAO1, PEG, PLGA.
- Figure 2:
- Abbreviations in the caption:
- Many of them do not appear in the figure: AT, CASP2, DsiRNA, EGFR, ATTR, NC, LDL-C, NP, PS, TRPV1.
- Some of the pictures are similar to others previously published in reference 6 (DOI: 10.1002/wrna.1662). As some of the authors are common to both papers I assume there is not a conflict.
- Abbreviations in the caption:
- The definition of some acronyms is missing:
- Line 359: BACH1, PICALM, JARID2. They are targets of miR-155 and although they are properly referenced (59) it would be advisable to add a brief description of how they contribute to patient survival.
- Line 363: HDAC
- Line 606: MPM instead of MRM for “Malignant pleural mesothelioma”.
- Line 673: There is no explanation about the LODER system mentioned.
Author Response
We are grateful for the insightful comments provided by the reviewers. Below we outline point-by-point responses to the Reviewers’ remarks. Our responses are in italics.
Reviewer Comments:
Reviewer #1
In the paper “Clinical applications of RNA-based therapies in the era of precision medicine” the authors show a review of current uses of siRNAs and/or miRNAs for clinical therapy.
RNA-based strategies for gene regulation are gaining interest and they have become a trending subject matter in the last years. Their use in molecular medicine research has provided a rationale for the therapeutical treatment of several genetic disorders.
As a consequence, a great number of reviews dealing with this topic have been recently published and it seems difficult to contribute to new advances in this field. Many of those reviews have been properly referenced by the authors and are focused in miRNA-based (3,6) or siRNA-based (22) therapies.
Fortunately, the authors have been able to detect and exploit gaps in the literature so they can offer a new viewpoint: they integrate both miRNA and siRNA-based therapies and they focus on those that have entered clinical trial evaluation or are FDA-approved.
Furthermore, the writing is ordered, clear and easy to understand. I am glad to observe that not only satisfactory results are presented but also those that have failed because they do not meet the expectations or those that have been prematurely terminated due to technical or medical incidences.
We appreciate the overall positive comments by the Reviewer. Below we provide a point-by-point response to the constructive critiques provided by the Reviewer.
- Increase the accuracy of the title: RNA-based therapies not only rely on short non-coding RNAs, but also in mRNAs, long non-coding RNAs, circular RNAs or ribozymes (see https://doi.org/10.1080/15476286.2022.2027150). It would be more appropriate to use “clinical applications of therapies based on sncRNAs…”, because the current title do not fulfill the expectances.
We are thankful to the Reviewer for this suggestion. We have accordingly modified the title: “Clinical Applications of short non-coding RNA-based Therapies in the Era of Precision Medicine”
- Figure 1:
- What do “IDT”and “Dicerna” stand for? Are they the names of the companies that provide the dsRNAs? It is confusing, as it seems to be a schematic representation of sRNA-mediated silencing in the cell.
We spell out these terms. They refer to two approaches to produce siRNAs.
- Abbreviations in the caption: Some of them are missing: LDHA. Some of them do not appear in the figure: HAO1, PEG, PLGA.
We have corrected this to eliminate any inconsistency between what is shown in the figure and the figure legend.
- Figure 2:
- Abbreviations in the caption: Many of them do not appear in the figure: AT, CASP2, DsiRNA, EGFR, ATTR, NC, LDL-C, NP, PS, TRPV1.
We have corrected this to eliminate any inconsistency between what is shown in the figure and the figure legend.
- Some of the pictures are similar to others previously published in reference 6 (DOI: 10.1002/wrna.1662). As some of the authors are common to both papers I assume there is not a conflict.
The Reviewer is correct. We have copyright and an open research license agreement to modify these figures files.
- The definition of some acronyms is missing:
- Line 359: BACH1, PICALM, JARID2. They are targets of miR-155 and although they are properly referenced (59) it would be advisable to add a brief description of how they contribute to patient survival.
We have added a description, as suggested.
- Line 363: HDAC
We spell this out.
- Line 606: MPM instead of MRM for “Malignant pleural mesothelioma”.
We have corrected this.
- Line 673: There is no explanation about the LODER system mentioned.
We spell out what LODER stands for.
Reviewer 2 Report
Abstract: Well written and fully describes the contents of the manuscript.
Introduction: Well written, covers basic comparative descriptions of two broad classes of ncRNA therapeutics (siRNA and miRNA therapies).
Figures and Figure Legends: The figures are beautiful and informative, however individual panels are not labeled (i.e. Panel A, B, etc.) This would help the reader when the figures are referenced in the text. For example, Figure 1 is referenced about 5 times in the text, but the reader has to infer from the context which image in Figure 1 is relevant to the section they are reading. Similarly it would be good to reference the individual panels in the figure in the figure legend, instead of having one long paragraph where the reader has to pick apart which sentences are referring to which image in the figure. If the authors could make this change to Figures 1 and 2 as well has the in-text references, it would greatly enhance the reading experience.
Section 2 - Delivery Strategies and Chemical Modifications: Good
Section 3 - Clinical Applications ncRNAs: excellent presentation covering both the therapeutic targets, current or historical clinical trials as well as their outcomes
Section 4 - miRNA therapeutics preclinical - good although the discussion on miR-21 talks about its potential as a therapeutic target and some complications in that respect, but doesn't detail any preclinical finding or preclinical work as included in discussions on the other targets in this section. Perhaps an extra sentence are two discussing some of the miR-21 preclincal work highlighted in the Bautista-Sanchez reference would add to value of this section.
Conclusion: excellent discussion on the remaining challenges in the field of RNA therapeutics and the future outlook for this field of medicine
Author Response
We are grateful for the insightful comments provided by the reviewers. Below we outline point-by-point responses to the Reviewers’ remarks. Our responses are in italics.
Reviewer Comments:
Reviewer #2
Abstract: Well written and fully describes the contents of the manuscript.
Introduction: Well written, covers basic comparative descriptions of two broad classes of ncRNA therapeutics (siRNA and miRNA therapies).
We appreciate the overall positive comments by the Reviewer. Below we provide a point-by-point response to the constructive critiques provided by the Reviewer.
- Figures and Figure Legends: The figures are beautiful and informative, however individual panels are not labeled (i.e. Panel A, B, etc.) This would help the reader when the figures are referenced in the text. For example, Figure 1 is referenced about 5 times in the text, but the reader has to infer from the context which image in Figure 1 is relevant to the section they are reading. Similarly it would be good to reference the individual panels in the figure in the figure legend, instead of having one long paragraph where the reader has to pick apart which sentences are referring to which image in the figure. If the authors could make this change to Figures 1 and 2 as well has the in-text references, it would greatly enhance the reading experience.
We have added panel labels as suggested by the Reviewer.
- Section 2 - Delivery Strategies and Chemical Modifications: Good
- Section 3 - Clinical Applications ncRNAs: excellent presentation covering both the therapeutic targets, current or historical clinical trials as well as their outcomes.
- Section 4 - miRNA therapeutics preclinical - good although the discussion on miR-21 talks about its potential as a therapeutic target and some complications in that respect, but doesn't detail any preclinical finding or preclinical work as included in discussions on the other targets in this section. Perhaps an extra sentence are two discussing some of the miR-21 preclinical work highlighted in the Bautista-Sanchez reference would add to value of this section.
We were careful to keep the review focused on miRNA studies in the clinical setting, since there are a lot of studies in the preclinical setting, which are beyond the scope of the review. However, we added a brief summary of preclinical work on miR-21 for more context.
- Conclusion: excellent discussion on the remaining challenges in the field of RNA therapeutics and the future outlook for this field of medicine.